# Spectrum of urolith composition among a multi-ethnic population at the Aga Khan hospital, Dar-es-Salaam, Tanzania

Mapendo Providence[1*], Hussam Uddin Soomro[1], Natasha Housseine[2], Ali Akbar Zehri[1]

1 Department of Surgery, The Aga Khan University-Medical College East Africa, Dar-es-Salaam, Tanzania,
2 Department of Obstetrics and Gynaecology, The Aga Khan University-Medical College East Africa, Dar-es-Salaam, Tanzania

* lucky_prov@yahoo.com

## Abstract

### Background

Urolithiasis is among the commonest diseases affecting the urinary tract with worldwide prevalence varying from 1%−20%. A urolith refers to a calculus or stone within the urinary tract, knowledge of urolith composition is important in understanding their etiology, treatment and preventing recurrence. This study aimed to describe the spectrum of urolith compositions among patients seen at the Aga Khan Hospital Dar-es-Salaam, Tanzania.

### Methods

This was a hospital based cross-sectional study carried out among patients with urolithiasis treated during a ten-year period from January 2011 to December 2020 whose stones were sent for stone analysis by infrared spectroscopy. Data on age, sex, clinical presentation, mode of diagnosis, mode of extraction and chemical composition was analyzed.

### Results

A total of 168 adult patients were included in the study, with a mean age of 44.7 yrs. (SD 10.99). Males (83.3%) were more affected than females. The majority (58.3%) of stones were found in the ureters. Purely calcium oxalate stones was the predominant composition of urinary tract stones, found in 66.1% of patients, this predominance was observed across both genders and across all age groups and anatomical locations, following calcium oxalate only stones in prevalence were stones with mixed calcium oxalate and calcium phosphate (21.4%).

**Data availability statement:** Data used in the study includes clinical data as well as socio-demographic data of participants involved in the study. It is property of the Aga Khan University - Dar-es-Salaam Campus (a third party). Authors in this study were affiliated to the university either as a student or faculty at the university and were granted permission to use this data by the institution's research committee via ethical clearance, after completion of the study the data set was submitted to the University's research office. This data can be available on reasonable request to the Aga Khan University- Dar-es-Salaam Campus' research administrative office. Correspondences can be made to Ms. Mwanaarab R. Sibuma from the research administrative office. Email: mwanaarab.sibuma@aku.edu

**Funding:** The author(s) received no specific funding for this work.

**Competing interests:** The authors declare that they have no competing interests.

## Conclusion

Stones in our setting were of mixed composition. The most common constituent of stones across all ages, gender and locations was calcium oxalate only stones. Male gender was most predominant and the commonest site of urinary calculi was the ureters.

## Introduction

Urolithiasis is one among the most common diseases affecting the urinary tract, third only to urinary tract infections and prostate pathologies [1]. Worldwide prevalence varies from 1–20% with higher prevalences of > 10% being found among populations with a higher standard of living [2]. In North America it ranges from 7–13%, 5–9% in Europe and 1–5% in Asia. Urolithiasis is thought to be less common in tropical Africa than in the western world [3].

In Tanzania data on true incidence and prevalence of urolithiasis is limited, a study by Mkony et al done in 1991 over a six month period estimated the incidence to be 243 per 100,000, an incidence rate that is comparable to that of the western world [4].

A urolith refers to a calculus or stone within the urinary tract, difference in incidence of these urinary tract stones depends on geographical, climatic, ethnic, dietary and genetic factors [2]. It also depends on socio-economic status, water consumption, water quality [5] and occupation [6]. Urolithiasis is generally more prevalent among males; studies suggest a gender ratio of 1.5–2.5 worldwide. In recent years, a narrowing gender gap has been evident which may be due to changes in diet and rise in metabolic syndrome [6,7]. Generally, incidence is lower in children and elderly patients, with peaks in the 4th to 6th decade of life. [6]

Urolithiasis is a highly recurrent disease [2,8] with at least 50% of individuals experiencing a recurrence of urinary stones within 10 years of the first occurrence [8].

Urinary tract stones can be classified based on their composition, [2]. Knowledge of stone composition helps in understanding pathophysiology, deciding modality of treatment and preventing recurrence of urolithiasis [9].

Urinary stone compositions are obtained from both chemical and physical methods of stone analysis. The European Association of Urology (EAU) recommends infrared spectroscopy and X-ray diffraction as the preferred procedures for analysis. [2].

Recommendations by both the American Urological Association (AUA) and European Association of Urology is that stone analysis be performed at least once in a patient with urolithiasis [2,10].

The most common component of urinary calculi is calcium found in approximately 75% of stones. Among calcium stones most are composed of calcium oxalate about 60%, mixed calcium oxalate and hydroxy apatite (calcium phosphate) 20%. Other common stone compositions are uric acid approximately 10%, struvite (ammonium magnesium phosphate) about 10%, brushite (a form of calcium phosphate) 2% and cysteine 1% [9].

Urinary stone composition is dependent on lifestyle and diet, which in turn depends on the country, climate and culture. Other factors which influence urinary stone composition include age, gender, and anatomical location of stone within the urinary tract

In Tanzania urolithiasis was more prevalent in males, males being affected 2–3 times more than females [11,12]. A previous study was done in Tanzania in 1991 by Mkony et al, among 44 patients with urinary stone disease, of which 20 stones were sent for analysis, 40% of patients were found to have calcium oxalate stones while 35% had calcium phosphate stones and 25% had mixed stones [4].

A gap exists in knowledge of the composition of urinary stones in our setting [6]. The last study done on composition of urinary tract stones was over thirty years ago and on a limited number of patients and with use of chemical methods that have now been considered inadequate [2,4,13]. Since then, there have been few studies on urinary tract stones in Tanzania and even among those none has been done to ascertain the composition of urinary tract calculi in our region [11,12].

Knowledge of urinary stone composition is important in the prevention and management of urinary tract stones. Most of the existing data on stone composition has been from western and developed countries. Due to existing differences in climate, dietary intake, ethnicity and socioeconomic factors between Tanzania and these countries, these data might not match the stone composition profile in our setting. The aim of this study was to describe the composition of urinary tract stones among patients in Dar-es-Salaam, Tanzania.

## Methods

The study design was a hospital based cross-sectional study among patients who underwent treatment for urinary tract stones during a 10 year period between January 2011 to December 2020 at the Aga Khan Hospital, Dar-es- Salaam, a private teaching and tertiary care referral hospital located in Dar-es-Salaam, Tanzania. The hospital has multiple specialties including Urology, a unit in the Surgery department which provides endourology services for patients with urolithiasis requiring such intervention. A radiology department capable of diagnostic services such as X-ray, ultrasound and computed tomography (CT) scans which can be used in the diagnosis of urinary calculi, the hospital also offers laboratory services at which stones obtained from patients are submitted and subsequently sent for analysis by infrared spectroscopy. The hospital attends both referral and self-referred patients and caters for a multi-ethnic population.

A consecutive list of urinary tract stones sent for stone analysis was obtained from the laboratory. Medical records of these patients were then traced and data on patients' socio-demographic characteristics, clinical presentation, presence of urinary tract infections, comorbidities, mode of stone diagnosis, and anatomical location of stone and method of extraction of stone were collected retrospectively from the patient medical records from 1st December 2022–30th March 2023. Data on stone composition determined by infrared spectroscopy was retrieved from the laboratory.

Descriptive statistics such as age, sex, ethnicity, comorbidities, clinical presentation, and location of urolith, mode of diagnosis, method of stone extraction and composition of uroliths were categorized and analyzed by frequencies, percentages, means and standard deviation (SD). For inferential statistics, differences in urolith composition found between groups categorized by age, sex and anatomical location of urolith were analyzed using chi square test and Fisher's exact test whereby a P-value of $< 0.05$ was considered to be statistically significant.

## Ethical approval

Before commencement of the study, ethical approval was sought from the ethics research committee of the Aga Khan University (AKU) with ethical committee approval number: AKU/2022/014/fb/011 and ethics committee approval date: 14th November 2022. Throughout the study confidentiality of patients was ensured by anonymization of patient related information. Patients were identified by their medical record numbers and assigned a serial number as their unique identifier in the data collection tool. Consent for participation was not applicable as it was a retrospective study.

## Results

The study included one hundred and sixty-eight participants, Majority (83.3%) of whom were male, with a male to female ratio of 5:1, age of participants ranged from 24 to 68 years of age, with mean age of 44.7 years (SD 10.99), most commonly affected age group was the 40–49 years age group category (30.4%) followed by 30–39 years age group category (28.6%). Majority of the patients were residing in Dar-es-Salaam (85.7%), Majority (61.3%) of the patients were Tanzanian, and ethnicities of participants were African, Asian and Caucasian ethnicity (Table 1).

Across all age groups, urinary calculi were more prevalent among male patients than female patients, most common age group with urinary calculi for men was the 30–39 years age group (32.1%) while in females most patients with urinary calculi were among the 40–49 yrs. age group (39.3%). In both genders patients who fell within the 20–29 years and 60–69 years age groups had the least occurrence of uroliths (Fig 1).

The most common clinical features on presentation were flank pain (94.4%) and hematuria (24.4%). Only a quarter of the patients (25.6%) were found to have features of urinary tract infections on urinalysis. Among all patients, the most frequently found comorbidities were hypertension and diabetes (Table 2).

Majority (78%) of patients at the time of presentation had a single urinary tract stone, the most common site (58.3%) of urinary calculi were the ureters followed by patients having stones lodged in both the kidneys and ureters (21.4%), bladder stones were not common (1.8%). All but one (99.4%) of the stones were diagnosed by CT scan, and the most common method of extracting the uroliths (63.7%) was ureteroscopy (URS) and lithoclast. All stones were analyzed by infrared spectroscopy (Table 3).

Across all anatomical sites of lodgement, urinary tract stones were more common in the male gender, Among males most common site of lodgement were the ureters, similarly among females. Bladder stones were only found among male patients (Fig 2).

Kidney stones were more common in the 30–39 age group (18.8%) and 50–59 age groups (20.5%),while ureteric stones were most common in the 40–49 age group (72.5%), bladder stones were found only in the 60–69 yrs age group, and constituted 16.7% of all stones in this age group (Fig 3).

**Table 1. Socio-demographic characteristics of participants.**

| Socio-demographic characteristics | |
|---|---|
| Variable | n = 168 |
| **Gender, n (%)** | |
| Male | 140 (83.3) |
| Female | 28 (16.7) |
| **Age categories, n (%)** | |
| 20–29 | 12 (7.1) |
| 30–39 | 48 (28.6) |
| 40–49 | 51 (30.4) |
| 50–59 | 39 (23.2) |
| 60–69 | 18(10.7) |
| **Residence, n (%)** | |
| Dar | 144 (85.7) |
| Other | 24 (14.3) |
| **Nationality, n (%)** | |
| Tanzanian | 103 (61.3) |
| Non-Tanzanian | 65 (38.7) |

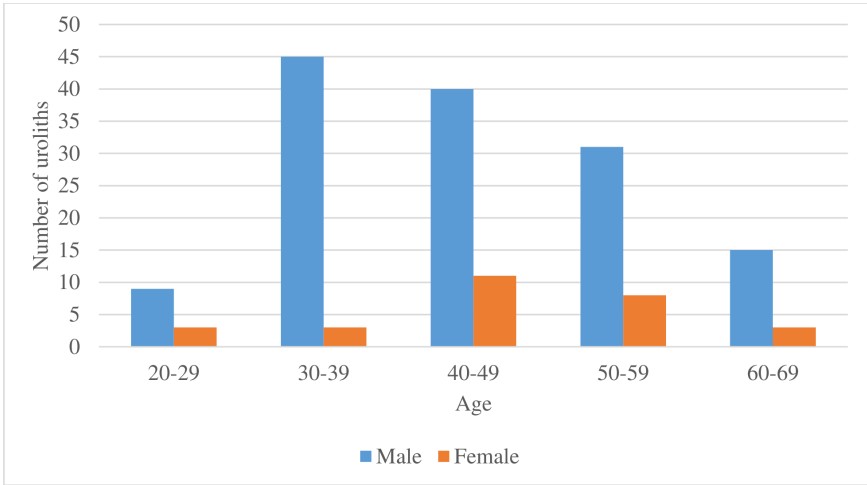

**Fig 1. Distribution of urinary calculi by age and gender.**

**Table 2. Clinical characteristics of patients with urolithiasis.**

| Clinical characteristics | |
|---|---|
| **Clinical presentation, n (%)** | |
| Flank pain | 159 (94.6) |
| Hematuria | 41 (24.4) |
| Fever | 3 (1.8) |
| Nausea | 14 (8.3) |
| Vomiting | 20 (11.9) |
| Painful urination | 20 (11.9) |
| **Co-existing urinary tract infection, n (%)** | |
| Yes | 43 (25.6) |
| No | 94 (56.0) |
| Missing | 31 (18.4) |
| **Comorbidities, n (%)** | |
| Hypertension | 31 (20.0) |
| Diabetes | 15 (9.7) |
| Hyperuricemia or gout | 4 (2.6) |
| Others | 25 (14.9) |
| None | 101 (60.0) |
| Missing | 13 (7.7) |

Ten different compostions of urinary calculi were identified. Calcium oxalate occured in 99.4% of the stones either alone or in combination with other compositions, only one stone did not contain calcium oxalate as a constituent. Majority (66.1%) were purely calcium oxalate stones, followed by mixed calcium oxalate and calcium phosphate stones (21.4%) and calcium oxalate and calcium oxalate and uric acid stones (6%). Mixed calcium and struvite (ammonium magnesium phospate) stones were only 2.4% while calcium oxalate and ammonium urate stones were 1.2%, the rest of the compositions were 0.6% each (Table 4).

**Table 3. Characteristics of urinary calculi.**

| Urinary calculi characteristics | |
| --- | --- |
| **Number of stones, n (%)** | |
| Single stone | 131 (78.0) |
| Multiple stones | 37 (22.0) |
| **Anatomical site of stones, n (%)** | |
| Kidney | 30 (17.9) |
| Ureter | 98 (58.3) |
| Bladder | 3 (1.8) |
| Kidney+ Ureter | 36 (21.4) |
| Ureter +Bladder | 1 (0.6) |
| **Imaging mode of diagnosis, n (%)** | |
| CT scan | 167 (99.4) |
| Ultrasound | 1 (0.6) |
| **Mode of stone extraction (%)** | |
| URS+ Laser lithoclast | 107 (63.7) |
| URS+ Laser lithotripsy | 53 (31.5) |
| Percutaneous nephrolithotomy (PCNL) | 4 (2.4) |
| Spontaneous passage | 4 (2.4) |

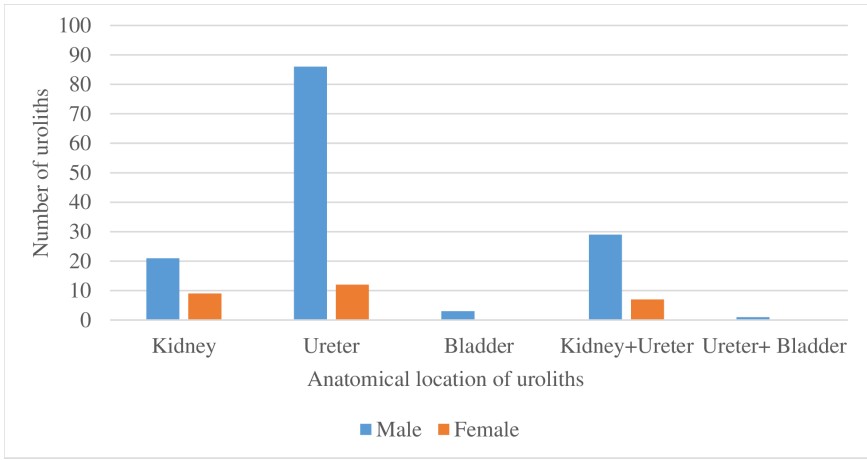

**Fig 2. Distribution of calculi by location and gender.**

Compositions of urinary calculi were similar across the genders for calcium oxalate only stones and mixed calcium oxalate and calcium phosphate stones. Mixed calcium oxalate and ammonium magnesium phosphate stone were found among only male patients, while calculi containing uric acid and urates were slightly more common in females (14.3%), than males (9.3%), but these differences were not found to be statistically significant (Table 5).

Calcium oxalate stones were found to be more in the 60–68 yrs age group with a slight tendency to decrease in proportion when moving through lower age groups. Among stones containing calcium phosphate proportion was highest among the 20–29 yrs and there was a slight tendency for proportion of calcium phosphate containing stone to decrease as age increase through the age groups but these differences were not found to be statistically significant all with p-values > 0.05.

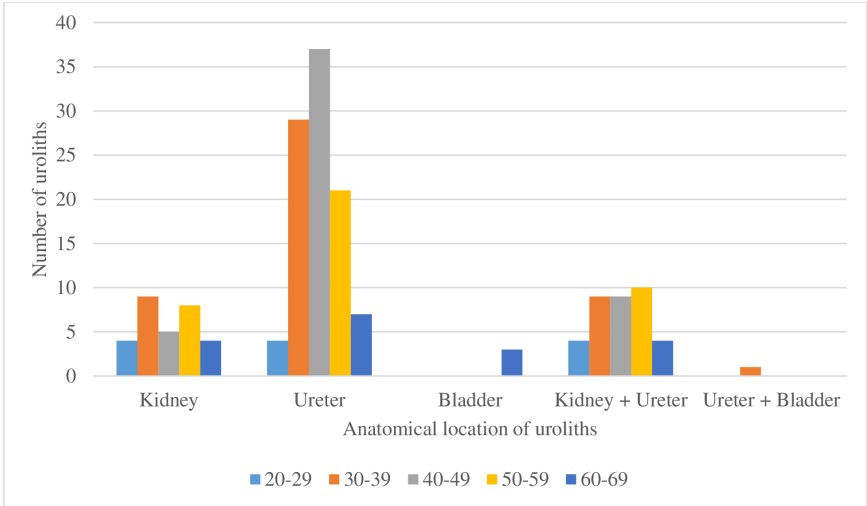

**Fig 3. Distribution of calculi by age and location.**

**Table 4. Composition of urinary calculi.**

| Composition of urinary calculi, n (%) | n = 168 |
|---|---|
| Calcium oxalate only | 111 (66.1) |
| Calcium oxalate + Calcium phosphate | 36 (21.4) |
| Calcium oxalate + Uric acid | 10 (6.0) |
| Calcium oxalate + Ammonium magnesium phosphate | 4 (2.4) |
| Calcium oxalate + Ammonium urate | 2 (1.2) |
| Calcium oxalate + Sodium urate | 1 (0.6) |
| Calcium oxalate + Calcium phosphate+ Uric acid | 1 (0.6) |
| Calcium oxalate + Calcium phosphate+ Ammonium urate | 1 (0.6) |
| Calcium oxalate+ Uric acid + Ammonium urate | 1 (0.6) |
| Uric acid + Sodium urate | 1 (0.6) |

**Table 5. Composition of urinary calculi by gender.**

| Composition of urinary calculi by gender | | | |
|---|---|---|---|
| Composition, n (%) | Gender | | |
| | Male, n = 140 | Female = 28 | P-value |
| Calcium oxalate only | 93 (66.4) | 18 (64.3) | 0.827 |
| Calcium oxalate + calcium phosphate | 30 (21.4) | 6 (21.4) | 1.000 |
| Calcium oxalate + Ammonium magnesium phosphate | 4 (2.9) | 0 (0.0) | 1.000 |
| Mixed composition uric acid and urate containing stones | 13 (9.3) | 4 (14.3) | 0.490 |

Among the stones containing ammonium magnesium phosphate all were found in the 50–59 yrs age group, while calculi containing uric acid and urates were found to have highest porportion in the 30–39 yrs age group (20.8%). These differences across age groups among these two categories were found to be statistically significant p = 0.019 and p = 0.047 respectively (Table 6).

**Table 6. Composition of urinary calculi by age groups.**

| Composition, n (%) | Age group categories | | | | | |
|---|---|---|---|---|---|---|
| | 20-29, n=12 | 30-39, n=48 | 40-49, n=51 | 50-59, n=39 | 60-68, n=18 | p-value |
| Calcium oxalate | 8 (66.7) | 27 (56.3) | 33 (64.7) | 29 (74.4) | 14 (77.8) | 0.371 |
| Calcium oxalate + Calcium phosphate | 4 (33.3) | 11 (22.9) | 14 (27.5) | 5 (12.8) | 2 (11.1) | 0.279 |
| Calcium oxalate + Ammonium magnesium phosphate | 0 (0.0) | 0 (0.0) | 0 (0.0) | 4 (10.3) | 0 (0.0) | 0.019 |
| Mixed composition uric acid and urate containing stones | 0 (0.0) | 10 (20.8) | 4 (7.8) | 1 (2.6) | 2 (11.1) | 0.047 |

Purely calcium oxalate stones were found to predominate across all anatomical locations, calcium phosphate containing stones were mostly found in the ureters, bladder stones were purely calcium oxalate, ammonium magnesium phosphate containing stones were only found in the ureters. On analysis by chi-square and Fisher's exact test, these observed differences were not statistically significant (Fig 4).

## Discussion

This study provides a description of urinary tract stones and their compositions in our setting. Urolithiasis was found to be a male predominant disease, mostly affecting those within the 4th decade of life. Majority of patients had a single urinary tract stone and the most common site of lodgement was the ureters. Calcium oxalate found alone or mixed with other consistuents was the most common composition across all ages, genders and anatomical sites.

A male to female ratio of 5:1 was found in the current study, similar to the study previously done in Tanzania and to studies from Kenya, Northern Nigeria and Ireland where male to female ratios were 3.4:1, 3.8:1, 6:1 and 1.9:1 respectively [3,4,14,15]. Higher levels of androgens which promote stone formation are among factors which explain this predominance in males [6,16].

Participant ages ranged from 24 to 68 years, most commonly affected was the 40–49 years age group (30.4%) followed by 30–39 years (28.6%). These results were similar to a study from South Eastern Nigeria which found the 30–49 years age group to have the highest (50%) number of urinary tract stones [17],The increased incidence in this age

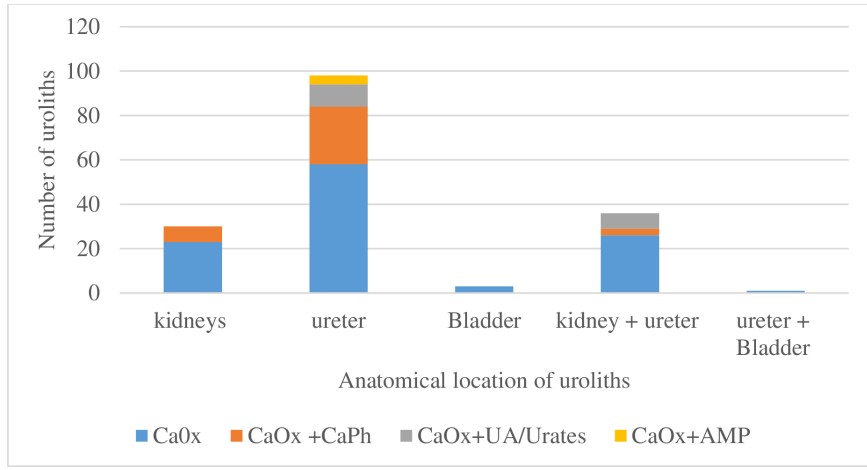

**Fig 4. Composition of urinary calculi by anatomical site.**

group could be attributable to dietary and life style changes. Also those in middle age are more prone to working outdoors for long durations thus increasing likelihood of dehydration and urinary stone formation especially in hot climates such as in Dar-es-Salaam where majority of the participants resided [6,16].

Majority of stones were found in the ureters and kidneys, similar to Kenya where majority (46.5%) of stones were found in the ureters followed by the pelvi-ureteric junction 25.4% and in Thailand were majority (66.5%) of the stones were renal stones and ureteric stones in 33.5% of patients [3,18]. In this study bladder stones were rare (1.8%), none of the patients had urethral stones, this was similar to studies from the western world but different from studies in Africa where bladder stones were more common. The previous Tanzanian study had a higher incidence of bladder stones and urethral stones (30%). [4,6,15,17].

Consistent with studies across the world, the major composition of urinary calculi was calcium oxalate, it was contained either alone or in combination with other stones in 99.4% of stones, only one stone did not contain calcium oxalate. This was similar to studies in Ireland and Nigeria where calcium oxalate was the major composition of stones in 94.5% and 93.5% of stones respectively. We found 66.1% purely calcium oxalate stones which was similar to studies in Morrocco 66.6% and Kenya 71.6%. The second commonest type of uroliths were calcium phosphate stones (21.4%) followed by calcium oxalate and uric acid stones (6%). Ammonium magnesium phosphate stones only constituted 2.4% of stones [3,16,19].

The previous Tanzanian study found calcium oxalate in 40% of all stones, a prevalence lower than that observed in this current study. Proportion of calcium phosphate stones (35%) was also slightly higher. These differences reveal a change in urinary calculi composition in our country over the past 30 years, and suggest a difference in diet, lifestyle and socio-economic status of people over time. Also, the previous study was done in a public hospital while the current study setting was a private hospital which further reveals the role of socioeconomic status in stone composition as services in private hospitals tend to be more expensive compared to public hospitals [4].

No statistically significant differences were observed when comparing composition by gender, a finding which was similar to the 2017 study in Kenya [3]. All ammonium magnesium phosphate (struvite or infection) stones were only found in men, which is different from other studies where these stones are more common in women due to a higher risk of urinary tract infections arising from their shorter length of the urethra. Such as in China where prevalence of infection stones in their setting was twice more in women than men (17.22% versus 8.27%) [20]. Across all age groups, purely calcium oxalate stones predominated, proportion of calcium phosphate stone decreased with increasing age but these differences were not noted to be statistically significant, p = 0.285.

A statistically significant difference existed in composition of stones across age groups, ammonium magnesium phosphate stones were found in age 50–59 years age group (10.3%) and among male patients only. These stones are thought to be common in persons over 50 years of age, because they are more prone to urinary tract infections [15,17]. But as the number of participants with ammonium magnesium phosphate (struvite) stones is this study was very low other studies with a greater number of participants having this particular stone compostion is needed inorder to confirm this finding.

No statistically significant differences were found in composition of stones by anatomical site of lodgement. Other studies reported this difference especially among bladder stones. Such as a Nigerian study in which 85.8% bladder stones were struvite composition, showing association of bladder stones with urinary tract infection [15,17].

## Limitations

This was a single center study with participants predominantly residing in Dar-es-Salaam region. It was done in a private hospital where demographics including socio-economic status could be different from patients attending public hospitals. Also, a selection bias as only stone compositions of patients whose stones were submitted to the lab for analysis were studied. Hence, results obtained may not be generalized to the whole population. Lastly this was a retrospective study

hence other factors which could influence stone composition such as diet, duration of residence in a specific location, and incomplete medical records data on comorbidities and urinary tract infections could not be studied retrospectively.

## Conclusion

Urinary tract stones in our setting were of mixed composition. The most common constituent of stones across all ages, gender and location were calcium oxalate only stones. Male gender was most predominant and common site of urinary calculi were the ureters. There were no statistically significant differences observed in the composition of urinary calculi across genders. A statistically significant difference was observed among stones containing Ammonium magnesium phosphate and those containing uric acid & urate containing stone with age, but a larger study is recommended to confirm this observation due to a low number of participants with both stone compositions in this study.

## Supporting information

**S1. Appendix**
(DOCX)

## Acknowledgments

Our thanks to the hospital management and the medical records staff at the Aga Khan Hospital, Dar-es-Salaam for their support and cooperation.

## Author contributions

**Conceptualization:** Mapendo Providence, Ali Akbar Zehri.

**Data curation:** Mapendo Providence, Ali Akbar Zehri.

**Formal analysis:** Mapendo Providence, Hussam Uddin Soomro, Natasha Housseine, Ali Akbar Zehri.

**Investigation:** Mapendo Providence.

**Methodology:** Mapendo Providence, Natasha Housseine.

**Supervision:** Ali Akbar Zehri.

**Writing – original draft:** Mapendo Providence, Hussam Uddin Soomro, Natasha Housseine, Ali Akbar Zehri.

**Writing – review & editing:** Mapendo Providence, Ali Akbar Zehri.

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
