## [Decision Letter · Decision Letter 0]

22 Nov 2024

PONE-D-24-31701Spectrum of urolith composition among a multi-ethnic population at the Aga Khan Hospital, Dar-es-Salaam, Tanzania.PLOS ONE

Dear Dr. Providence,

Thank you for submitting your manuscript to PLOS ONE. After careful consideration, we feel that it has merit but does not fully meet PLOS ONE’s publication criteria as it currently stands. Therefore, we invite you to submit a revised version of the manuscript that addresses the points raised during the review process.

We look forward to receiving your revised manuscript.

Kind regards,

Engy Asem Ashaat

Academic Editor

PLOS ONE

Journal Requirements:

2. We note that your Data Availability Statement is currently as follows: “All relevant data are within the manuscript and in Supporting Information files.”

Please confirm at this time whether or not your submission contains all raw data required to replicate the results of your study. Authors must share the “minimal data set” for their submission. PLOS defines the minimal data set to consist of the data required to replicate all study findings reported in the article, as well as related metadata and methods (https://journals.plos.org/plosone/s/data-availability#loc-minimal-data-set-definition). For example, authors should submit the following data: - The values behind the means, standard deviations and other measures reported; - The values used to build graphs; - The points extracted from images for analysis. Authors do not need to submit their entire data set if only a portion of the data was used in the reported study. If your submission does not contain these data, please either upload them as Supporting Information files or deposit them to a stable, public repository and provide us with the relevant URLs, DOIs, or accession numbers. For a list of recommended repositories, please see https://journals.plos.org/plosone/s/recommended-repositories. If there are ethical or legal restrictions on sharing a de-identified data set, please explain them in detail (e.g., data contain potentially sensitive information, data are owned by a third-party organization, etc.) and who has imposed them (e.g., an ethics committee). Please also provide contact information for a data access committee, ethics committee, or other institutional body to which data requests may be sent. If data are owned by a third party, please indicate how others may request data access.

3. Please ensure that you refer to Figures 1-4 in your text as, if accepted, production will need this reference to link the reader to the figure.

4. We note you have included a table to which you do not refer in the text of your manuscript. Please ensure that you refer to Table 1-6 in your text; if accepted, production will need this reference to link the reader to the Table.

Reviewers' comments:

Reviewer's Responses to Questions

**Comments to the Author**

1. Is the manuscript technically sound, and do the data support the conclusions?

Reviewer #1: Yes

Reviewer #2: Partly

2. Has the statistical analysis been performed appropriately and rigorously? 

Reviewer #1: Yes

Reviewer #2: Yes

3. Have the authors made all data underlying the findings in their manuscript fully available?

Reviewer #1: Yes

Reviewer #2: No

4. Is the manuscript presented in an intelligible fashion and written in standard English?

Reviewer #1: Yes

Reviewer #2: Yes

5. Review Comments to the Author

Reviewer #1: The study presented in the manuscript makes a valuable contribution to the field. The authors have conducted a thorough investigation and have addressed the research question with appropriate methodology and analysis. The data analysis is sound, and the findings are well-supported by the provided evidence.

Comments to the authors:

• In the manuscript title do you mean AGA KHAN HOSPITAL Please check the spelling all over the manuscript.

• In the introduction “ Since then, there have been few studies on urinary tract stones in Tanzania and even among those none has been done to ascertain the composition of urinary tract calculi in our region..” Please cite the references of those few studies if possible.

Reviewer #2: The topic of the study is not novel and many factors should be studied in detailed manner including family history of chronic kidney disease,special dietary habits, cerebrovascular ds ,special medications

6. PLOS authors have the option to publish the peer review history of their article (what does this mean? ). If published, this will include your full peer review and any attached files.

**Do you want your identity to be public for this peer review?** For information about this choice, including consent withdrawal, please see our Privacy Policy .

Reviewer #1: **Yes: ** Fatma Mohamed Abdelmordy

Reviewer #2: **Yes: ** Seham A. Elazab

---

## [Author Response · Author response to Decision Letter 1]

24 Jan 2025

Response to reviewer #1's comments.

1st Comment :In the manuscript title do you mean Aga Khan Hospital. Please check spelling all over the manuscript.

Response : Thank you for the correction. Yes, I meant the Aga Khan Hospital. Spelling has been corrected form ‘hopital’ to hospital, and the spelling checked throughout the document.

2nd Comment: In the introduction, “since then there have been few studies on urinary tract stones in Tanzania and even among those none have been done to ascertain the composition of urinary tract calculi in our region”. Please cite the references of those few studies if possible.

Response: References for these studies have been cited.

Reviewer #2's comments.

Comment: The topic of the study is not novel and many factors should be studied in detailed manner including family history of chronic kidney disease special dietary habits, special medications etc.

Response: The main objective of our study was to describe the compositions of urinary tract stones in our region. We also studied a larger number of patients compared to other studies that had been previously done in our setting.

Being a retrospective study obtaining data on the mentioned factors was not possible due to inadequate information from the patients’ medical records, and this was addressed by being included as a limitation to our study.

---

## [Decision Letter · Decision Letter 1]

24 Apr 2025

PONE-D-24-31701R1Spectrum of urolith composition among a multi-ethnic population at the Aga Khan Hospital, Dar-es-Salaam, Tanzania.PLOS ONE

Dear Dr. Providence,

Thank you for submitting your manuscript to PLOS ONE. After careful consideration, we feel that it has merit but does not fully meet PLOS ONE’s publication criteria as it currently stands. Therefore, we invite you to submit a revised version of the manuscript that addresses the points raised during the review process.

We look forward to receiving your revised manuscript.

Kind regards,

Murat Akand, MD, PhD, FEBU

Academic Editor

PLOS ONE

Journal Requirements:

Additional Editor Comments:

Thank you for the revised version of your manuscript. Although various points have been appropriately addressed within this revision, I see there are still some more points to be corrected appropriately:

1. "Urolith" is not a commonly used term. Therefore, I recommend that the authors use "stone disease", "kidney stone", or something similar according to the context of the sentence in which this word is used.

2. There are numerous spelling and punctuation errors to be corrected: for example, leaving a space before the parenthesis or a comma, starting with capital words for some words, such as Ultrasound, Computed Tomography, or writing with small letters instead of capittal letters, such as American urological association, European urological association, etc. Please correct these kinds of errors.

3. Please start the Discussion section with a summary of your results. Please also avoid mentioning the same general information that has already been mentioned in the Introduction section.

4. Please give the ethics committee approval date and number.

5. Please place the tables and figures (along with their legends) separately at the end of the manuscript. Placing them in the main text makes the reading of the manuscript broken and more difficult.

6. Please leave a space between the numbers (n) and the percentages (%) in the tables.

7. It might be better to reorganize the stone types in Table 4, as calcium oxalate + combined with other types can be merged; at least the ones other than calcium oxalate + calcium phosphate and calcium oxalate + uric acid, as the other mixed types are less common.

Reviewers' comments:

Reviewer's Responses to Questions

**Comments to the Author**

1. If the authors have adequately addressed your comments raised in a previous round of review and you feel that this manuscript is now acceptable for publication, you may indicate that here to bypass the “Comments to the Author” section, enter your conflict of interest statement in the “Confidential to Editor” section, and submit your "Accept" recommendation.

Reviewer #3: All comments have been addressed

2. Is the manuscript technically sound, and do the data support the conclusions?

Reviewer #3: Yes

3. Has the statistical analysis been performed appropriately and rigorously? 

Reviewer #3: Yes

4. Have the authors made all data underlying the findings in their manuscript fully available?

Reviewer #3: Yes

5. Is the manuscript presented in an intelligible fashion and written in standard English?

Reviewer #3: Yes

6. Review Comments to the Author

Reviewer #3: The revised manuscript was good written and discussed.

All comments of reviewers were addressed in the manuscript.

7. PLOS authors have the option to publish the peer review history of their article (what does this mean? ). If published, this will include your full peer review and any attached files.

**Do you want your identity to be public for this peer review?** For information about this choice, including consent withdrawal, please see our Privacy Policy .

Reviewer #3: No

---

## [Author Response · Author response to Decision Letter 2]

29 Jun 2025

Comment:

Response: The reference list has been reviewed. One reference by Jung et al was removed from the reference list after revisions were made to the discussion section of the manuscript and paragraphs in which it was cited removed, these revisions were as per the academic editor’s comments to remove general information from this section that is similar to that already mentioned in the introduction section.

Below is the full citation for the omitted reference:

Jung H, Andonian S, Assimos D, Averch T, Geavlete P, Kohjimoto Y, et al. Urolithiasis: evaluation, dietary factors, and medical management: an update of the 2014 SIU-ICUD international consultation on stone disease. World journal of urology. 2017;35(9):1331-40.

Academic editor’s comments.

1. “Urolith” is not a commonly used term. Therefore, I recommend that the authors use “stone disease”, “kidney stone” or something similar according to the context of the sentence in which this word is used.

Response:

We have reduced the use of the word "urolith" within most of the manuscript and have replaced it with either the words "urinary tract stones", "stones", "urinary calculi" or "calculi" depending on the sentence in which it was used. We have also added a definition of the term urolith within the introduction section. We agree that " urolith" is not a commonly used term however there exists a number of literature which use the word urolith to describe urinary tract stones, as such we request to retain the use of the word "urolith" especially within the title.

2.There are numerous spelling and punctuation errors to be corrected: for example, leaving a space before the parenthesis or a comma, starting with capital words for some words, such as Ultrasound, Computed Tomography, or writing with small letters instead of capital letters, such as American urological association, European urological association, etc. Please correct these kinds of errors.

Response:

Spelling and punctuation errors have been corrected throughout the manuscript.

3.Please start the Discussion section with a summary of your results. Please also avoid mentioning the same general information that has already been mentioned in the Introduction section.

Response:

The discussion section has been revised. It has been started with a summary of the results and general information already mentioned in the introduction section has been removed.

4.Please give the ethics committee approval date and number.

Response:

Ethics committee approval date and number have been provided in the methods section of the manuscript. They have also been added to the ethical statement section included in the submission process.

5.Please place the tables and figures (along with their legends) separately at the end of the manuscript. Placing them in the main text makes the reading of the manuscript broken and more difficult.

Response:

Figure captions and tables were placed within the manuscript as this was in accordance to PLOS ONE’s submission guidelines.

The guidelines state that, "Figure captions must be inserted in the text of the manuscript, immediately following the paragraph in which the figure is first cited (read order). Do not include captions as part of the figure files themselves or submit them in a separate document". Regarding tables they state, "Place each table in your manuscript file directly after the paragraph in which it is first cited (read order). Do not submit your tables in separate files." Kindly let us know if we did not understand or follow these guidelines correctly so that we can rectify accordingly.

6.Please leave a space between the numbers (n) and the percentages (%) in the tables.

Response:

A space has been placed between the numbers and the percentages in all the tables in the manuscript.

7.It might be better to reorganize the stone types in Table 4, as calcium oxalate + combines with other types can be merged, at least the ones other than calcium oxalate + calcium phosphate and calcium oxalate + uric acid, as other mixed types are less common.

Response:

As our main objective was to describe the whole spectrum of composition of urinary tracts stones that we studied, we kindly request to maintain the categories of compositions of urinary tract stones in table 4, so as to better describe the whole variety of stones found in our study.

Reviewer #3’s comment.

Comment: The revised manuscript was good written and discussed. All comments of reviewers were addressed in the manuscript.

Response: Thank you for feedback and also for the time and effort that you have dedicated to our manuscript.

---

## [Editor Report · Decision Letter 2]

18 Jul 2025

Spectrum of urolith composition among a multi-ethnic population at the Aga Khan Hospital, Dar-es-Salaam, Tanzania.

PONE-D-24-31701R2

Dear Dr. Providence,

We’re pleased to inform you that your manuscript has been judged scientifically suitable for publication and will be formally accepted for publication once it meets all outstanding technical requirements.

Kind regards,

Murat Akand, MD, PhD, FEBU

Academic Editor

PLOS ONE
---

## [Editor Report · Acceptance letter]

PONE-D-24-31701R2

PLOS ONE

Dear Dr. Providence,

I'm pleased to inform you that your manuscript has been deemed suitable for publication in PLOS ONE. Congratulations! Your manuscript is now being handed over to our production team.

Kind regards,

on behalf of

Dr. Murat Akand

Academic Editor

PLOS ONE